# Generalizing Hamiltonian Monte Carlo with Neural Networks

**Daniel Levy**[1]*, **Matthew D. Hoffman**[2], **Jascha Sohl-Dickstein**[3]
[1]Stanford University, [2]Google AI Perception , [3]Google Brain
`danilevy@cs.stanford.edu`, `{mhoffman,jaschasd}@google.com`

## Abstract

We present a general-purpose method to train Markov chain Monte Carlo kernels, parameterized by deep neural networks, that converge and mix quickly to their target distribution. Our method generalizes Hamiltonian Monte Carlo and is trained to maximize expected squared jumped distance, a proxy for mixing speed. We demonstrate large empirical gains on a collection of simple but challenging distributions, for instance achieving a $106\times$ improvement in effective sample size in one case, and mixing when standard HMC makes no measurable progress in a second. Finally, we show quantitative and qualitative gains on a real-world task: latent-variable generative modeling. We release an open source TensorFlow implementation of the algorithm.

## 1 Introduction

High-dimensional distributions that are only analytically tractable up to a normalizing constant are ubiquitous in many fields. For instance, they arise in protein folding (Schütte et al., 1999), physics simulations (Olsson, 1995), and machine learning (Andrieu et al., 2003). Sampling from such distributions is a critical task for learning and inference (MacKay, 2003), however it is an extremely hard problem in general.

Markov Chain Monte Carlo (MCMC) methods promise a solution to this problem. They operate by generating a sequence of correlated samples that converge in distribution to the target. This convergence is most often guaranteed through detailed balance, a sufficient condition for the chain to have the target equilibrium distribution. In practice, for any proposal distribution, one can ensure detailed balance through a Metropolis-Hastings (Hastings, 1970) accept/reject step.

Despite theoretical guarantees of eventual convergence, in practice convergence and mixing speed depend strongly on choosing a proposal that works well for the task at hand. What's more, it is often more art than science to know when an MCMC chain has converged ("burned-in"), and when the chain has produced a new uncorrelated sample ("mixed"). Additionally, the reliance on detailed balance, which assigns equal probability to the forward and reverse transitions, often encourages random-walk behavior and thus slows exploration of the space (Ichiki & Ohzeki, 2013).

For densities over continuous spaces, Hamiltonian Monte Carlo (HMC; Duane et al., 1987; Neal, 2011) introduces independent, auxiliary momentum variables, and computes a new state by integrating Hamiltonian dynamics. This method can traverse long distances in state space with a single Metropolis-Hastings test. This is the state-of-the-art method for sampling in many domains. However, HMC can perform poorly in a number of settings. While HMC mixes quickly spatially, it struggles at mixing across energy levels due to its volume-preserving dynamics. HMC also does not work well with multi-modal distributions, as the probability of sampling a large enough momentum to traverse a very low-density region is negligibly small. Furthermore, HMC struggles with ill-conditioned energy landscapes (Girolami & Calderhead, 2011) and deals poorly with rapidly changing gradients (Sohl-Dickstein et al., 2014).

Recently, probabilistic models parameterized by deep neural networks have achieved great success at *approximately* sampling from highly complex, multi-modal *empirical* distributions (Kingma &

---

*Work was done while the author was at Google Brain.

Welling, 2013; Rezende et al., 2014; Goodfellow et al., 2014; Bengio et al., 2014; Sohl-Dickstein et al., 2015). Building on these successes, we present a method that, given an *analytically* described distribution, automatically returns an *exact* sampler with good convergence and mixing properties, from a class of highly expressive parametric models. The proposed family of samplers is a generalization of HMC; it transforms the HMC trajectory using parametric functions (deep networks in our experiments), while retaining theoretical guarantees with a tractable Metropolis-Hastings accept/reject step. The sampler is trained to minimize a variation on expected squared jumped distance (similar in spirit to Pasarica & Gelman (2010)). Our parameterization reduces easily to standard HMC. It is further capable of emulating several common extensions of HMC such as within-trajectory tempering (Neal, 1996) and diagonal mass matrices (Bennett, 1975).

We evaluate our method on distributions where HMC usually struggles, as well as on a the real-world task of training latent-variable generative models.

Our contributions are as follows:

- We introduce a generic training procedure which takes as input a distribution defined by an energy function, and returns a fast-mixing MCMC kernel.

- We show significant empirical gains on various distributions where HMC performs poorly.

- We finally evaluate our method on the real-world task of training and sampling from a latent variable generative model, where we show improvement in the model's log-likelihood, and greater complexity in the distribution of posterior samples.

## 2 RELATED WORK

Adaptively modifying proposal distributions to improve convergence and mixing has been explored in the past (Andrieu & Thoms, 2008). In the case of HMC, prior work has reduced the need to choose step size (Neal, 2011) or number of leapfrog steps (Hoffman & Gelman, 2014) by adaptively tuning those parameters. Salimans et al. (2015) proposed an alternate scheme based on variational inference. We adopt the much simpler approach of Pasarica & Gelman (2010), who show that choosing the hyperparameters of a proposal distribution to maximize expected squared jumped distance is both principled and effective in practice.

Previous work has also explored applying models from machine learning to MCMC tasks. Kernel methods have been used both for learning a proposal distribution (Sejdinovic et al., 2014) and for approximating the gradient of the energy (Strathmann et al., 2015). In physics, Restricted and semi-Restricted Boltzmann machines have been used both to build approximations of the energy function which allow more rapid sampling (Liu et al., 2017; Huang & Wang, 2017), and to motivate new hand-designed proposals (Wang, 2017).

Most similar to our approach is recent work from Song et al. (2017), which uses adversarial training of a volume-preserving transformation, which is subsequently used as an MCMC proposal distribution. While promising, this technique has several limitations. It does not use gradient information, which is often crucial to maintaining high acceptance rates, especially in high dimensions. It also can only indirectly measure the quality of the generated sample using adversarial training, which is notoriously unstable, suffers from "mode collapse" (where only a portion of a target distribution is covered), and often requires objective modification to train in practice (Arjovsky et al., 2017). Finally, since the proposal transformation preserves volume, it can suffer from the same difficulties in mixing across energy levels as HMC, as we illustrate in Section 5.

To compute the Metropolis-Hastings acceptance probability for a deterministic transition, the operator must be invertible and have a tractable Jacobian. Recent work (Dinh et al., 2016), introduces RNVP, an invertible transformation that operates by, at each layer, modifying only a subset of the variables by a function that depends solely on the remaining variables. This is exactly invertible with an efficiently computable Jacobian. Furthermore, by chaining enough of these layers, the model can be made arbitrarily expressive. This parameterization will directly motivate our extension of the leapfrog integrator in HMC.

## 3 BACKGROUND

### 3.1 MCMC METHODS AND METROPOLIS-HASTINGS

Let $p$ be a target distribution, analytically known up to a constant, over a space $\mathcal{X}$. Markov chain Monte Carlo (MCMC) methods (Neal, 1993) aim to provide samples from $p$. To that end, MCMC methods construct a Markov Chain whose stationary distribution is the target distribution $p$. Obtaining samples then corresponds to simulating a Markov Chain, i.e., given an initial distribution $\pi_0$ and a transition kernel $K$, constructing the following sequence of random variables:

$$X_0 \sim \pi_0, \quad X_{t+1} \sim K(\cdot|X_t). \tag{1}$$

In order for $p$ to be the stationary distribution of the chain, three conditions must be satisfied: $K$ must be irreducible and aperiodic (these are usually mild technical conditions) and $p$ has to be a fixed point of $K$. This last condition can be expressed as: $p(x') = \int K(x'|x)p(x)\mathrm{d}x$. This condition is most often satisfied by satisfying the stronger *detailed balance* condition, which can be written as: $p(x')K(x|x') = p(x)K(x'|x)$.

Given any proposal distribution $q$, satisfying mild conditions, we can easily construct a transition kernel that respects detailed balance using Metropolis-Hastings (Hastings, 1970) accept/reject rules. More formally, starting from $x_0 \sim \pi_0$, at each step $t$, we sample $x' \sim q(\cdot|X_t)$, and with probability $A(x'|x_t) = \min\left(1, \frac{p(x')q(x_t|x')}{p(x_t)q(x'|x_t)}\right)$, accept $x'$ as the next sample $x_{t+1}$ in the chain. If we reject $x'$, then we retain the previous state and $x_{t+1} = x_t$. For typical proposals this algorithm has strong asymptotic guarantees. But in practice one must often choose between very low acceptance probabilities and very cautious proposals, both of which lead to slow mixing. For continuous state spaces, Hamiltonian Monte Carlo (HMC; Neal, 2011) tackles this problem by proposing updates that move far in state space while staying roughly on iso-probability contours of $p$.

### 3.2 HAMILTONIAN MONTE CARLO

Without loss of generality, we assume $p(x)$ to be defined by an energy function $U(x)$, s.t. $p(x) \propto \exp(-U(x))$, and where the state $x \in \mathbb{R}^n$. HMC extends the state space with an additional momentum vector $v \in \mathbb{R}^n$, where $v$ is distributed independently from $x$, as $p(v) \propto \exp(-\frac{1}{2}v^T v)$ (i.e., identity-covariance Gaussian). From an augmented state $\xi \triangleq (x, v)$, HMC produces a proposed state $\xi' = (x', v')$ by approximately integrating Hamiltonian dynamics jointly on $x$ and $v$, with $U(x)$ taken to be the potential energy, and $\frac{1}{2}v^T v$ the kinetic energy. Since Hamiltonian dynamics conserve the total energy of a system, their approximate integration moves along approximate iso-probability contours of $p(x, v) = p(x)p(v)$.

The dynamics are typically simulated using the leapfrog integrator (Hairer et al., 2003; Leimkuhler & Reich, 2004), which for a single time step consists of:

$$v^{\frac{1}{2}} = v - \tfrac{\epsilon}{2}\partial_x U(x); \quad x' = x + \epsilon v^{\frac{1}{2}}; \quad v' = v - \tfrac{\epsilon}{2}\partial_x U(x'). \tag{2}$$

Following Sohl-Dickstein et al. (2014), we write the action of the leapfrog integrator in terms of an operator $\mathbf{L}$: $\mathbf{L}\xi \triangleq \mathbf{L}(x, v) \triangleq (x', v')$, and introduce a momentum flip operator $\mathbf{F}$: $\mathbf{F}(x, v) \triangleq (x, -v)$. It is important to note two properties of these operators. First, the transformation $\mathbf{FL}$ is an involution, i.e. $\mathbf{FLFL}(x, v) = \mathbf{FL}(x', -v') = (x, v)$. Second, the transformations from $(x, v)$ to $(x, v^{\frac{1}{2}})$, from $(x, v^{\frac{1}{2}})$ to $(x', v^{\frac{1}{2}})$, and from $(x', v^{\frac{1}{2}})$ to $(x', v')$ are all volume-preserving *shear* transformations i.e., only one of the variables ($x$ or $v$) changes, by an amount determined by the other one. The determinant of the Jacobian, $\left|\frac{\partial[\mathbf{FL}\xi]}{\partial\xi^T}\right|$, is thus easy to compute. For vanilla HMC $\left|\frac{\partial[\mathbf{FL}\xi]}{\partial\xi^T}\right| = 1$, but we will leave it in symbolic form for use in Section 4. The Metropolis-Hastings-Green (Hastings, 1970; Green, 1995) acceptance probability for the HMC proposal is made simple by these two properties, and is

$$A(\mathbf{FL}\xi|\xi) = \min\left(1, \frac{p(\mathbf{FL}\xi)}{p(\xi)}\left|\frac{\partial[\mathbf{FL}\xi]}{\partial\xi^T}\right|\right). \tag{3}$$

# 4 L2HMC: TRAINING MCMC SAMPLERS

In this section, we describe our proposed method L2HMC (for 'Learning To Hamiltonian Monte Carlo'). Given access to only an energy function $U$ (and not samples), L2HMC learns a parametric leapfrog operator $\mathbf{L}_\theta$ over an augmented state space. We begin by describing what desiderata we have for $\mathbf{L}_\theta$, then go into detail on how we parameterize our sampler. Finally, we conclude this section by describing our training procedure.

## 4.1 AUGMENTING HMC

HMC is a powerful algorithm, but it can still struggle even on very simple problems. For example, a two-dimensional multivariate Gaussian with an ill-conditioned covariance matrix can take arbitrarily long to traverse (even if the covariance is diagonal), whereas it is trivial to sample directly from it. Another problem is that HMC can only move between energy levels via a random walk (Neal, 2011), which leads to slow mixing in some models. Finally, HMC cannot easily traverse low-density zones. For example, given a simple Gaussian mixture model, HMC cannot mix between modes without recourse to additional tricks, as illustrated in Figure 1b. These observations determine the list of desiderata for our learned MCMC kernel: *fast mixing*, *fast burn-in*, *mixing across energy levels*, and *mixing between modes*.

While pursuing these goals, we must take care to ensure that our proposal operator retains two key features of the leapfrog operator used in HMC: it must be invertible, and the determinant of its Jacobian must be tractable. The leapfrog operator satisfies these properties by ensuring that each sub-update only affects a subset of the variables, and that no sub-update depends nonlinearly on any of the variables being updated. We are free to generalize the leapfrog operator in any way that preserves these properties. In particular, we are free to translate and rescale each sub-update of the leapfrog operator, so long as we are careful to ensure that these translation and scale terms do not depend on the variables being updated.

### 4.1.1 STATE SPACE

As in HMC, we begin by augmenting the current state $x \in \mathbb{R}^n$ with a continuous momentum variable $v \in \mathbb{R}^n$ drawn from a standard normal. We also introduce a binary direction variable $d \in \{-1, 1\}$, drawn from a uniform distribution. We will denote the complete augmented state as $\xi \triangleq (x, v, d)$, with probability density $p(\xi) = p(x)p(v)p(d)$. Finally, to each step $t$ of the operator $\mathbf{L}_\theta$ we assign a fixed random binary mask $m^t \in \{0, 1\}^n$ that will determine which variables are affected by each sub-update. We draw $m^t$ uniformly from the set of binary vectors satisfying $\sum_{i=1}^n m_i^t = \lfloor \frac{n}{2} \rfloor$, that is, half of the entries of $m^t$ are 0 and half are 1. For convenience, we write $\bar{m}^t = \mathbb{1} - m^t$ and $x_{m^t} = x \odot m^t$ ($\odot$ denotes element-wise multiplication, and $\mathbb{1}$ the all ones vector).

### 4.1.2 UPDATE STEPS

We now describe the details of our augmented leapfrog integrator $\mathbf{L}_\theta$, for a single time-step $t$, and for direction $d = 1$.

We first update the momenta $v$. This update can only depend on a subset $\zeta_1 \triangleq (x, \partial_x U(x), t)$ of the full state, which excludes $v$. It takes the form

$$v' = v \odot \exp(\tfrac{\epsilon}{2} S_v(\zeta_1)) - \tfrac{\epsilon}{2} \left( \partial_x U(x) \odot \exp(\epsilon Q_v(\zeta_1)) + T_v(\zeta_1) \right). \tag{4}$$

We have introduced three new functions of $\zeta_1$: $T_v$, $Q_v$, and $S_v$. $T_v$ is a translation, $\exp(Q_v)$ rescales the gradient, and $\exp(\tfrac{\epsilon}{2} S_v)$ rescales the momentum. The determinant of the Jacobian of this transformation is $\exp\left(\tfrac{\epsilon}{2} \mathbb{1} \cdot S_v(\zeta_1)\right)$. Note that if $T_v$, $Q_v$, and $S_v$ are all zero, then we recover the standard leapfrog momentum update.

We now update $x$. As hinted above, to make our transformation more expressive, we first update a subset of the coordinates of $x$, followed by the complementary subset. The first update, which yields $x'$ and affects only $x_{m^t}$, depends on the state subset $\zeta_2 \triangleq (x_{\bar{m}^t}, v, t)$. Conversely, with $x'$ defined below, the second update only affects $x'_{\bar{m}^t}$ and depends only on $\zeta_3 \triangleq (x'_{m^t}, v, t)$:

$$\begin{aligned} x' &= x_{\bar{m}^t} + m^t \odot [x \odot \exp(\epsilon S_x(\zeta_2)) + \epsilon(v' \odot \exp(\epsilon Q_x(\zeta_2)) + T_x(\zeta_2))] \\ x'' &= x'_{m^t} + \bar{m}^t \odot [x' \odot \exp(\epsilon S_x(\zeta_3)) + \epsilon(v' \odot \exp(\epsilon Q_x(\zeta_3)) + T_x(\zeta_3))] . \end{aligned} \tag{5}$$

Again, $T_x$ is a translation, $\exp(Q_x)$ rescales the effect of the momenta, $\exp(\epsilon S_x)$ rescales the positions $x$, and we recover the original leapfrog position update if $T_x = Q_x = S_x = 0$. The determinant of the Jacobian of the first transformation is $\exp(\epsilon m^t \cdot S_x(\zeta_2))$, and the determinant of the Jacobian of the second transformation is $\exp(\epsilon \bar{m}^t \cdot S_x(\zeta_3))$.

Finally, we update $v$ again, based on the subset $\zeta_4 \triangleq (x'', \partial_x U(x''), t)$:

$$v'' = v' \odot \exp(\tfrac{\epsilon}{2} S_v(\zeta_4)) - \tfrac{\epsilon}{2}(\partial_x U(x'') \odot \exp(\epsilon Q_v(\zeta_4)) + T_v(\zeta_4)). \qquad (6)$$

This update has the same form as the momentum update in equation 4.

To give intuition into these terms, the scaling applied to the momentum can enable, among other things, acceleration in low-density zones, to facilitate mixing between modes. The scaling term applied to the gradient of the energy may allow better conditioning of the energy landscape (e.g., by learning a diagonal inertia tensor), or partial ignoring of the energy gradient for rapidly oscillating energies.

The corresponding integrator for $d = -1$ is given in Appendix A; it essentially just inverts the updates in equations 4, 5 and 6. For all experiments, the functions $Q, S, T$ are implemented using multi-layer perceptrons, with shared weights. We encode the current time step in the MLP input.

Our leapfrog operator $\mathbf{L}_\theta$ corresponds to running $M$ steps of this modified leapfrog, $\mathbf{L}_\theta \xi = \mathbf{L}_\theta(x, v, d) = (x''^{\times M}, v''^{\times M}, d)$, and our flip operator $\mathbf{F}$ reverses the direction variable $d$, $\mathbf{F}\xi = (x, v, -d)$. Written in terms of these modified operators, our proposal and acceptance probability are identical to those for standard HMC. Note, however, that this parameterization enables learning non-volume-preserving transformations, as the determinant of the Jacobian is a function of $S_x$ and $S_v$ that does not necessarily evaluate to 1. This quantity is derived in Appendix B.

### 4.1.3 MCMC Transitions

For convenience, we denote by $\mathbf{R}$ an operator that re-samples the momentum and direction. I.e., given $\xi = (x, v, d)$, $\mathbf{R}\xi = (x, v', d')$ where $v' \sim \mathcal{N}(0, I), d' \sim \mathcal{U}(\{-1, 1\})$. Sampling thus consists of alternating application of the $\mathbf{FL}_\theta$ and $\mathbf{R}$, in the following two steps each of which is a Markov transition that satisfies detailed balance with respect to $p$:

1. $\xi' = \mathbf{FL}_\theta \xi$ with probability $A(\mathbf{FL}_\theta \xi | \xi)$ (Equation 3), otherwise $\xi' = \xi$.
2. $\xi' = \mathbf{R}\xi$

This parameterization is effectively a generalization of standard HMC as it is non-volume-preserving, with learnable parameters, and easily reduces to standard HMC for $Q, S, T = 0$.

## 4.2 Loss and Training Procedure

We need some criterion to train the parameters $\theta$ that control the functions $Q$, $S$, and $T$. We choose a loss designed to reduce mixing time. Specifically, we aim to minimize lag-one autocorrelation. This is equivalent to maximizing expected squared jumped distance (Pasarica & Gelman, 2010). For $\xi, \xi'$ in the extended state space, we define $\delta(\xi', \xi) = \delta((x', v', d'), (x, v, d)) = ||x - x'||_2^2$. Expected squared jumped distance is thus $\mathbb{E}_{\xi \sim p(\xi)}[\delta(\mathbf{FL}_\theta \xi, \xi) A(\mathbf{FL}_\theta \xi | \xi)]$. However, this loss need not encourage mixing across the entire state space. Indeed, maximizing this objective can lead to regions of state space where almost no mixing occurs, so long as the average squared distance traversed remains high. To optimize both for typical and worst case behavior, we include a reciprocal term in the loss,

$$\ell_\lambda(\xi, \xi', A(\xi'|\xi)) = \frac{\lambda^2}{\delta(\xi, \xi') A(\xi'|\xi)} - \frac{\delta(\xi, \xi') A(\xi'|\xi)}{\lambda^2}, \qquad (7)$$

where $\lambda$ is a scale parameter, capturing the characteristic length scale of the problem. The second term encourages typical moves to be large, while the first term strongly penalizes the sampler if it is ever in a state where it cannot move effectively – $\delta(\xi, \xi')$ being small resulting in a large loss value. We train our sampler by minimizing this loss over both the target distribution and initialization distribution. Formally, given an initial distribution $\pi_0$ over $\mathcal{X}$, we define $q(\xi) = \pi_0(x)\mathcal{N}(v; 0, I)p(d)$, and minimize

$$\mathcal{L}(\theta) \triangleq \mathbb{E}_{p(\xi)}[\ell_\lambda(\xi, \mathbf{FL}_\theta \xi, A(\mathbf{FL}_\theta \xi | \xi))] + \lambda_b \mathbb{E}_{q(\xi)}[\ell_\lambda(\xi, \mathbf{FL}_\theta \xi, A(\mathbf{FL}_\theta \xi | \xi))]. \qquad (8)$$

The first term of this loss encourages mixing as it considers our operator applied on draws from the distribution; the second term rewards fast burn-in; $\lambda_b$ controls the strength of the 'burn-in' regularization. Given this loss, we exactly describe our training procedure in Algorithm 1. It is important to note that each training iteration can be done with only one pass through the network and can be efficiently batched. We further emphasize that this training procedure can be applied to any learnable operator whose Jacobian's determinant is tractable, making it a general framework for training MCMC proposals.

---

**Algorithm 1** Training L2HMC

---

**Input:** Energy function $U : \mathcal{X} \to \mathbb{R}$ and its gradient $\nabla_x U : \mathcal{X} \to \mathcal{X}$, initial distribution over the augmented state space $q$, number of iterations $n_{\text{iters}}$, number of leapfrogs $M$, learning rate schedule $(\alpha_t)_{t \leq n_{\text{iters}}}$, batch size $N$, scale parameter $\lambda$ and regularization strength $\lambda_b$.

Initialize the parameters of the sampler $\theta$.

Initialize $\{\xi_p^{(i)}\}_{i \leq N}$ from $q(\xi)$.

**for** $t = 0$ **to** $n_{\text{iters}} - 1$ **do**

    Sample a minibatch $\{\xi_q^{(i)}\}_{i \leq N}$ from $q(\xi)$.

    $\mathcal{L} \leftarrow 0$

    **for** $i = 1$ **to** $N$ **do**

        $\xi_p^{(i)} \leftarrow \mathbf{R}\xi_p^{(i)}$

        $\mathcal{L} \leftarrow \mathcal{L} + \ell_\lambda\left(\xi_p^{(i)}, \mathbf{FL}_\theta \xi_p^{(i)}, A(\mathbf{FL}_\theta \xi_p^{(i)}|\xi_p^{(i)})\right) + \lambda_b \ell_\lambda\left(\xi_q^{(i)}, \mathbf{FL}_\theta \xi_q^{(i)}, A(\mathbf{FL}_\theta \xi_q^{(i)}|\xi_q^{(i)})\right)$

        $\xi_p^{(i)} \leftarrow \mathbf{FL}_\theta \xi_p^{(i)}$ with probability $A(\mathbf{FL}_\theta \xi_p^{(i)}|\xi_p^{(i)})$.

    **end for**

    $\theta \leftarrow \theta - \alpha_t \nabla_\theta \mathcal{L}$

**end for**

---

## 5 Experiments

We present an empirical evaluation of our trained sampler on a diverse set of energy functions. We first present results on a collection of toy distributions capturing common pathologies of energy landscapes, followed by results on a task from machine learning: maximum-likelihood training of deep generative models. For each, we compare against HMC with well-tuned step length and show significant gains in mixing time. Code implementing our algorithm is available online[1].

### 5.1 Varied Collection of Energy Functions

We evaluate our L2HMC sampler on a diverse collection of energy functions, each posing different challenges for standard HMC.

**Ill-Conditioned Gaussian** (ICG): Gaussian distribution with diagonal covariance spaced log-linearly between $10^{-2}$ and $10^2$. This demonstrates that L2HMC can learn a diagonal inertia tensor.

**Strongly correlated Gaussian** (SCG): We rotate a diagonal Gaussian with variances $[10^2, 10^{-2}]$ by $\frac{\pi}{4}$. This is an extreme version of an example from Neal (2011). This problem shows that, although our parametric sampler only applies element-wise transformations, it can adapt to structure which is not axis-aligned.

**Mixture of Gaussians** (MoG): Mixture of two isotropic Gaussians with $\sigma^2 = 0.1$, and centroids separated by distance 4. The means are thus about 12 standard deviations apart, making it almost impossible for HMC to mix between modes.

**Rough Well**: Similar to an example from Sohl-Dickstein et al. (2014), for a given $\eta > 0, U(x) = \frac{1}{2}x^T x + \eta \sum_i \cos(\frac{x_i}{\eta})$. For small $\eta$ the energy itself is altered negligibly, but its gradient is perturbed by a high frequency noise oscillating between $-1$ and $1$. In our experiments, we choose $\eta = 10^{-2}$.

For each of these distributions, we compare against HMC with the same number of leapfrog steps and a well-tuned step-size. To compare mixing time, we plot auto-correlation for each method and

---

[1] https://github.com/brain-research/l2hmc

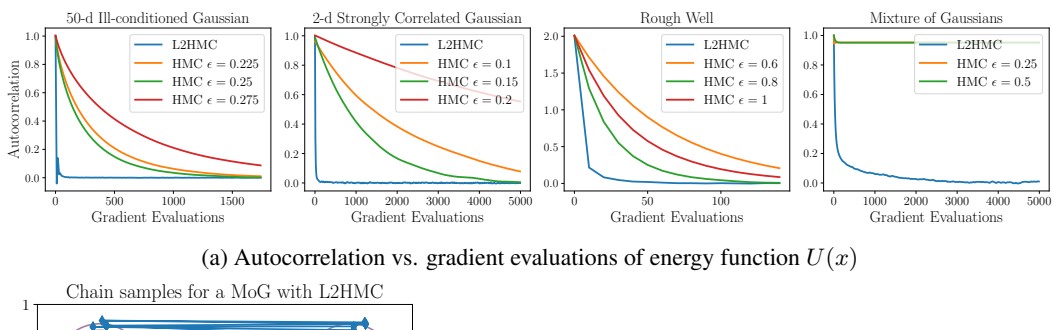

(a) Autocorrelation vs. gradient evaluations of energy function $U(x)$

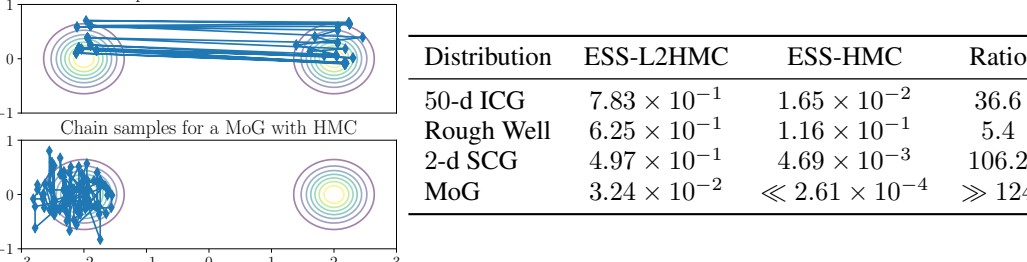

| Distribution | ESS-L2HMC | ESS-HMC | Ratio |
|---|---|---|---|
| 50-d ICG | $7.83 \times 10^{-1}$ | $1.65 \times 10^{-2}$ | 36.6 |
| Rough Well | $6.25 \times 10^{-1}$ | $1.16 \times 10^{-1}$ | 5.4 |
| 2-d SCG | $4.97 \times 10^{-1}$ | $4.69 \times 10^{-3}$ | 106.2 |
| MoG | $3.24 \times 10^{-2}$ | $\ll 2.61 \times 10^{-4}$ | $\gg 124$ |

(b) Samples from single MCMC chain     (c) ESS per Metropolis-Hastings step

(d) L2HMC can mix between modes for a MoG with different variances, contrary to A-NICE-MC.

Figure 1: L2HMC mixes faster than well-tuned HMC, and than A-NICE-MC, on a collection of toy distributions.

report effective sample size (ESS). We compute those quantities in the same way as Sohl-Dickstein et al. (2014). We observe that samplers trained with L2HMC show greatly improved autocorrelation and ESS on the presented tasks, providing more than $106\times$ improved ESS on the SCG task. In addition, for the MoG, we show that L2HMC can easily mix between modes while standard HMC gets stuck in a mode, unable to traverse the low density zone. Experimental details, as well as a comparison with LAHMC (Sohl-Dickstein et al., 2014), are shown in Appendix C.

**Comparison to A-NICE-MC (Song et al., 2017)** In addition to the well known challenges associated with adversarial training (Arjovsky et al., 2017), we note that parameterization using a volume-preserving operator can dramatically fail on simple energy landscapes. We build off of the *mog2* experiment presented in (Song et al., 2017), which is a 2-d mixture of isotropic Gaussians separated by a distance of 10 with variances 0.5. We consider that setup but increase the ratio of variances: $\sigma_1^2 = 3, \sigma_2^2 = 0.05$. We show in Figure 1d sample chains trained with L2HMC and A-NICE-MC; A-NICE-MC cannot effectively mix between the two modes as only a fraction of the volume of the large mode can be mapped to the small one, making it highly improbable to traverse. This is also an issue for HMC. On the other hand, L2HMC can both traverse the low-density region between modes, and map a larger volume in the left mode to a smaller volume in the right mode. It is important to note that the distance between both clusters is less than in the *mog2* case, and it is thus a good diagnostic of the shortcomings of volume-preserving transformations.

## 5.2 LATENT-VARIABLE GENERATIVE MODEL

We apply our learned sampler to the task of training, and sampling from the posterior of, a latent-variable generative model. The model consists of a latent variable $z \sim p(z)$, where we choose $p(z) = \mathcal{N}(z; 0, I)$, and a conditional distribution $p(x|z)$ which generates the image $x$. Given a family of parametric 'decoders' $\{z \mapsto p(x|z; \phi), \phi \in \Phi\}$, and a set of samples $\mathcal{D} = \{x^{(i)}\}_{i \leq N}$,

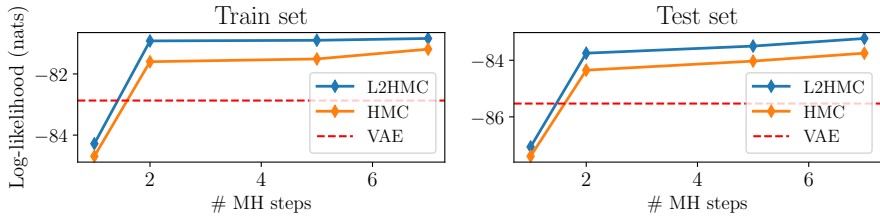

Figure 2: Training and held-out log-likelihood for models trained with L2HMC, HMC, and the ELBO (VAE).

training involves finding $\phi^* = \arg\max_{\phi \in \Phi} p(\mathcal{D}; \phi)$. However, the log-likelihood is intractable as $p(x; \phi) = \int p(x|z; \phi)p(z)\mathrm{d}z$. To remedy that problem, Kingma & Welling (2013) proposed jointly training an approximate posterior $q_\psi$ that maximizes a tractable lower-bound on the log-likelihood:

$$\mathcal{L}_{\text{ELBO}}(x, \phi, \psi) = \mathbb{E}_{q_\psi(z|x)}\left[p(x|z; \phi)\right] - \text{KL}(q_\psi(z|x)||p(z)) \leq p(x), \tag{9}$$

where $q_\psi(z|x)$ is a tractable conditional distribution with parameters $\psi$, typically parameterized by a neural network. Recently, to improve upon well-known pitfalls like over-pruning (Burda et al., 2015) of the VAE, Hoffman (2017) proposed HMC-DLGM. For a data sample $x^{(i)}$, after obtaining a sample from the approximate posterior $q_\psi(\cdot|x^{(i)})$, Hoffman (2017) runs a MCMC algorithm with energy function $U(z, x^{(i)}) = -\log p(z) - \log p(x^{(i)}|z; \phi)$ to obtain a more exact posterior sample from $p(z|x^{(i)}; \phi)$. Given that better posterior sample $z'$, the algorithm maximizes $\log p(x^{(i)}|z'; \phi)$.

To show the benefits of L2HMC, we borrow the method from Hoffman (2017), but replace HMC by jointly training an L2HMC sampler to improve the efficiency of the posterior sampling. We call this model **L2HMC-DLGM**. A diagram of our model and a formal description of our training procedure are presented in Appendix D. We define, for $\xi = \{z, v, d\}, r(\xi|x; \psi) \triangleq q_\psi(z|x)\mathcal{N}(v; 0, I)\mathcal{U}(d; \{-1, 1\})$.

In the subsequent sections, we compare our method to the standard VAE model from Kingma & Welling (2013) and HMC-DGLM from Hoffman (2017). It is important to note that, since our sampler is trained jointly with $p_\phi$ and $q_\psi$, *it performs exactly the same number of gradient computations of the energy function as HMC*. We first show that training a latent variable generative model with L2HMC results in better generative models both qualitatively and quantitatively. We then show that our improved sampler enables a more expressive, non-Gaussian, posterior.

**Implementation details:** Our decoder ($p_\phi$) is a neural network with 2 fully connected layers, with 1024 units each and softplus non-linearities, and outputs Bernoulli activation probabilities for each pixel. The encoder ($q_\psi$) has the same architecture, returning mean and variance for the approximate posterior. Our model was trained for 300 epochs with Adam (Kingma & Ba, 2014) and a learning rate $\alpha = 10^{-3}$. All experiments were done on the dynamically binarized MNIST dataset (LeCun).

### 5.2.1 SAMPLE QUALITY AND DATA LIKELIHOOD

We first present samples from decoders trained with L2HMC, HMC and the ELBO (i.e. vanilla VAE). Although higher log likelihood does not necessarily correspond to better samples (Theis et al., 2015), we can see in Figure 5, shown in the Appendix, that the decoder trained with L2HMC generates sharper samples than the compared methods.

We now compare our method to HMC in terms of log-likelihood of the data. As we previously stated, the marginal likelihood of a data point $x \in \mathcal{X}$ is not tractable as it requires integrating $p(x, z)$ over a high-dimensional space. However, we can estimate it using annealed importance sampling (AIS; Neal (2001)). Following Wu et al. (2016), we evaluate our generative models on both training and held-out data. In Figure 2, we plot the data's log-likelihood against the number of gradient computation steps for both HMC-DGLM and L2HMC-DGLM. We can see that for a similar number of gradient computations, L2HMC-DGLM achieves higher likelihood for both training and held-out data. This is a strong indication that L2HMC provides *significantly better posterior samples*.

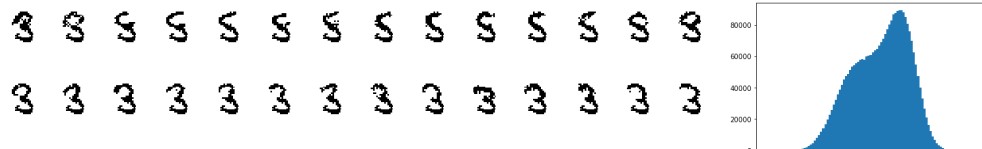

(a) Block Gibbs inpainting of the top half of an MNIST digit, using *(top)* L2HMC as a posterior sampler, and *(bottom)* $q_\psi$ as a posterior sampler.

(b) Non-Gaussian posterior

Figure 3: Demonstrations of the value of a more expressive posterior approximation.

### 5.2.2 INCREASED EXPRESSIVITY OF THE POSTERIOR

In the standard VAE framework, approximate posteriors are often parametrized by a Gaussian, thus making a strong assumption of uni-modality. In this section, we show that using L2HMC to sample from the posterior enables learning of a richer posterior landscape.

**Block Gibbs Sampling** To highlight our ability to capture more expressive posteriors, we in-paint the top of an image using Block Gibbs Sampling using the approximate posterior or L2HMC. Formally, let $x_0$ be the starting image. We denote top or bottom-half pixels as $x_0^{\text{top}}$ and $x_0^{\text{bottom}}$. At each step $t$, we sample $z^{(t)} \sim p(z|x_t; \theta)$, sample $\tilde{x} \sim p(x|z_t; \theta)$. We then set $x_{t+1}^{\text{top}} = \tilde{x}^{\text{top}}$ and $x_{t+1}^{\text{bottom}} = x_0^{\text{bottom}}$. We compare the results obtained by sampling from $p(z|x; \theta)$ using $q_\psi$ (i.e. the approximate posterior) vs. our trained sampler. The results are reported in Figure 3a. We can see that L2HMC easily mixes between modes (3, 5, 8, and plausibly 9 in the figure) while the approximate posterior gets stuck on the same reconstructed digit (3 in the figure).

**Visualization of the posterior** After training a decoder with L2HMC, we randomly choose an element $x_0 \in \mathcal{D}$ and run $512$ parallel L2HMC chains for $20,000$ Metropolis-Hastings steps. We then find the direction of highest variance, project the samples along that direction and show a histogram in Figure 3b. This plot shows non-Gaussianity in the latent space for the posterior. Using our improved sampler enables the decoder to make use of a more expressive posterior, and enables the encoder to sample from this non-Gaussian posterior.

## 6 FUTURE WORK

The loss in Section 4.2 targets lag-one autocorrelation. It should be possible to extend this to also target lag-two and higher autocorrelations. It should also be possible to extend this loss to reward fast decay in the autocorrelation of other statistics of the samples, for instance the sample energy as well as the sample position. These additional statistics could also include learned statistics of the samples, combining benefits of the adversarial approach of (Song et al., 2017) with the current work.

Our learned generalization of HMC should prove complementary to several other research directions related to HMC. It would be interesting to explore combining our work with the use of HMC in a minibatch setting (Chen et al., 2014); with shadow Hamiltonians (Izaguirre & Hampton, 2004); with gradient pre-conditioning approaches similar to those used in Riemannian HMC (Girolami et al., 2009; Betancourt, 2013); with the use of alternative HMC accept-reject rules (Sohl-Dickstein et al., 2014; Berger et al., 2015); with the use of non-canonical Hamiltonian dynamics (Tripuraneni et al., 2016); with variants of AIS adapted to HMC proposals (Sohl-Dickstein & Culpepper, 2012); with the extension of HMC to discrete state spaces (Zhang et al., 2012); and with the use of alternative Hamiltonian integrators (Creutz & Gocksch, 1989; Chao et al., 2015).

Finally, our work is also complementary to other methods not utilizing gradient information. For example, we could incorporate the intuition behind Multiple Try Metropolis schemes (Martino & Read, 2013) by having several parametric operators and training each one when used. In addition, one could draw inspiration from the adaptive literature (Haario et al., 2001; Andrieu & Thoms, 2008) or component-wise strategies (Gilks & Wild, 1992).

## 7 CONCLUSION

In this work, we presented a general method to train expressive MCMC kernels parameterized with deep neural networks. Given a target distribution $p$, analytically known up to a constant, our method provides a fast-mixing sampler, able to efficiently explore the state space. Our hope is that our method can be utilized in a "black-box" manner, in domains where sampling constitutes a huge bottleneck such as protein foldings (Schütte et al., 1999) or physics simulations (Olsson, 1995).

### ACKNOWLEDGMENTS

We would like to thank Ben Poole, Aditya Grover, David Belanger, and Colin Raffel for insightful comments on the draft, Mohammad Norouzi for support and encouragement launching the project, and Jiaming Song for discussions and help running A-NICE-MC.

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

# Appendix

## A  REVERSE LEAPFROG OPERATOR

Let $\xi = \{x, v, d\}$ in the extended state space with $d = -1$. Here, we describe the leapfrog updates for a single time step $t$, this consists of inverting the equations presented in the corresponding section.

Let $\zeta_1 = \{x, v, t\}$, we have:

$$v' = \left[v + \frac{\epsilon}{2} \left(\partial_x U(x) \odot \exp(\epsilon Q_v(\zeta_1)) + T_v(\zeta_1)\right)\right] \odot \exp(-S_v(\zeta_1)). \tag{10}$$

With the notation from Section 4, let $\zeta_2 \triangleq \{x_{m^t}, v, t\}$

$$x' = x_{m^t} + \bar{m}^t \odot \left[(x - \epsilon(\exp(\epsilon Q_x(\zeta_2)) \odot v' + T_x(\zeta_2))\right] \odot \exp(-\epsilon S_v(\zeta_2)). \tag{11}$$

Let us denote $\zeta_3 \triangleq \{x'_{\bar{m}^t}, v, t\}$:

$$x'' = x_{\bar{m}^t} + m^t \odot \left[(x' - \epsilon(\exp(\epsilon Q_x(\zeta_3)) \odot v' + T_x(\zeta_3))\right] \odot \exp(-\epsilon S_v(\zeta_3)). \tag{12}$$

Finally, the last update, with $\zeta_4 \triangleq \{x'', \partial_x U(x''), t\}$:

$$v' = \left[v + \frac{\epsilon}{2} \left(\partial_x U(x'') \odot \exp(\epsilon Q_v(\zeta_4)) + T_v(\zeta_4)\right)\right] \odot \exp(-S_v(\zeta_4)). \tag{13}$$

It is important to note that to invert $\mathbf{L}_\theta$, these steps should be ran for $t$ from $M$ to $1$.

## B  DETERMINANT OF THE JACOBIAN

Given the derivations (and notations) from Section 4, for the forward operator $\mathbf{L}_\theta$, we can immediately compute the Jacobian:

$$\log \left| \frac{\partial [\mathbf{FL}_\theta \xi]}{\partial \xi^T} \right| = d \sum_{t \leq M} \left[\frac{\epsilon}{2} \mathbf{1} \cdot S_v(\zeta_1^t) + \epsilon m^t \cdot S_x(\zeta_2^t) + \epsilon \bar{m}^t \cdot S_x(\zeta_3^t) + \frac{\epsilon}{2} \mathbf{1} \cdot S_v(\zeta_4^t)\right]. \tag{14}$$

Where $\zeta_i^t$ denotes the intermediary variable $\zeta_i$ at time step $t$ and $d$ is the direction of $\xi$ i.e. $\xi = \{x, v, d\}$.

## C  EXPERIMENTAL DETAILS OF SECTION 5

### C.1  IMPLEMENTATION DETAILS

First of all, we keep separate parameters for the network responsible for updating $v$ and those updating $x$. The architectures are the same. Let us take the example of $Q_v, S_v, T_v$. The time step $t$ is given as input to the MLP, encoded as $\tau(t) = \left(\cos(\frac{2\pi t}{M}), \sin(\frac{2\pi t}{M})\right)$. $\sigma(\cdot)$ denotes the ReLU non-linearity.

For $n_h$ hidden units per layer:

- We first compute $h_1 = \sigma(W_1 x + W_2 v + W_3 \tau(t) + b)$ ($h \in \mathbb{R}^{n_h}$).
- $h_2 = \sigma(W_4 h + b_4) \in \mathbb{R}^{n_h}$
- $S_v = \lambda_s \mathtt{tanh}(W_s h_2 + b_s), Q_v = \lambda_q \mathtt{tanh}(W_q h_2 + b_q), T_v = W_t h_2 + b_t$.

In Section 5.1, the $Q, S, T$ are neural networks with 2 hidden layers with 10 (100 for the 50-d ICG) units and ReLU non-linearities. We train with Adam (Kingma & Ba, 2014) and a learning rate $\alpha = 10^{-3}$. We train for $5,000$ iterations with a batch size of 200.

$\lambda_b$ was set to 0 for ICG and SCG and to 1 for MoG and Rough Well. For the MoG tasks, we train our sampler with a temperature parameter that we continuously anneal; we evaluate the trained sampler without using temperature.

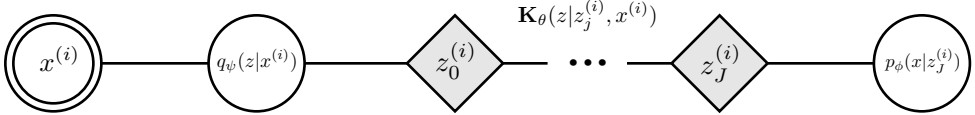

Figure 4: Diagram of our L2HMC-DGLM model. Nodes are functions of their parents. Round nodes are deterministic, diamond nodes are stochastic and the doubly-circled node is observed.

## C.2 Auto-correlation and ESS

Let $(x_\tau)_{\tau \leq T}$ be a set of correlated samples converging to the distribution $p$ with mean $\mu$ and covariance $\Sigma$. We define auto-correlation at time $t$ as:

$$\rho_t \triangleq \frac{1}{\text{Trace}(\Sigma)(T-t)} \sum_{\tau \leq T-t-1} (x_\tau - \mu)^T (x_{\tau+t} - \mu). \tag{15}$$

We can now define effective sample size (ESS) as:

$$\text{ESS}\left((x_\tau)_{\tau \leq T}\right) \triangleq \frac{1}{1 + 2\sum_t \rho_t}. \tag{16}$$

Similar to Hoffman & Gelman (2014), we truncate the sum when the auto-correlation goes below 0.05.

## C.3 Comparison with LAHMC

We compare our trained sampler with LAHMC (Sohl-Dickstein et al., 2014). Results are reported in Table 1. L2HMC largely outperforms LAHMC on all task. LAHMC is also *unable to mix between modes* for the MoG task. We also note that L2HMC could be easily combined with LAHMC, by replacing the leapfrog integrator of LAHMC with the learned one of L2HMC.

| Distribution | Gradient Evaluations | ESS-L2HMC | ESS-LAHMC | Ratio |
| --- | --- | --- | --- | --- |
| 50-d ICG | 2000 | 156.6 | 21.4 | 7.3 |
| Rough Well | 200 | 12.5 | 8.6 | 1.5 |
| 2-d SCG | 5000 | 116 | 16.7 | 14.9 |
| MoG | 20, 000 | 65.0 | $\ll 0.53$ | $\gg 123.5$ |

Table 1: ESS for a fixed number of gradient evaluations.

## D  L2HMC-DGLM

### D.1  Training algorithm

In this section, we present our training algorithm as well as a diagram explaining L2HMC-DGLM. For conciseness, given our operator $\mathbf{L}_\theta$, we denote by $\mathbf{K}_\theta(\cdot|x)$ the distribution over next state given sampling of a momentum and direction and the Metropolis-Hastings step.

### D.2  Implementation details of L2HMC-DGLM

Similar to our L2HMC training on unconditional sampling, we share weights across $Q, S$ and $T$. In addition, the auxiliary variable $x$ (here the image from MNIST) is first passed through a 2-layer neural network, with softplus non-linearities and $512$ hidden units. This input is given to both

---

**Algorithm 2** L2HMC for latent variable generative models

---

**Input:** dataset $\mathcal{D}$, number of iterations $n_{\text{iters}}$, number of Metropolis-Hastings step $J$, number of leapfrogs $M$, and learning rate schedule $(\alpha_t)_{t \leq n_{\text{iters}}}$.
Randomly initialize the decoder's parameters $\phi$ and the approximate posterior $\psi$. Initialize the parameters of the sampler $\theta$ with $M$ leapfrog steps.
**for** $t = 0$ **to** $n_{\text{iters}} - 1$ **do**
    Randomly sample a minibatch $\mathcal{B}$ from the dataset $\mathcal{D}$.
    $\mathcal{L}_{\text{ELBO}}, \mathcal{L}_{\text{Sampler}}, \mathcal{L}_{\text{Decoder}} \leftarrow 0$
    **for** $x^{(b)} \in \mathcal{B}$ **do**
        Sample $\xi_0^{(b)} \sim r(\cdot|x^{(b)}; \psi)$.
        $\mathcal{L}_{\text{ELBO}} \leftarrow p(x^{(b)}|z_0^{(b)}; \phi) - \text{KL}(q_\psi(z|x^{(b)})||p(z))$          ▷ With $\xi_0^{(b)} = \{z_0^{(b)}, v_0^{(b)}, d_0^{(b)}\}$
        Define the energy function $U_{x^{(b)}}(z) = -\log p(x^{(b)}|z; \theta) - \log p(z)$
        $\mathcal{L}_{\text{Sampler}} \leftarrow 0$
        $\lambda \leftarrow \sqrt{\text{Var}(q_\psi(z_0^{(b)}|x^{(b)})}$
        **for** $j = 0$ **to** $J - 1$ **do**
            $\xi_j^{(b)} \leftarrow \mathbf{R}\xi_j^{(b)}$
            $\mathcal{L}_{\text{Sampler}} \leftarrow \mathcal{L}_{\text{Sampler}} + \ell_\lambda(\xi_j^{(b)}, \mathbf{FL}_\theta\xi_j^{(b)}, A(\mathbf{FL}_\theta\xi_j^{(b)}|\xi_j^{(b)}))$
            Set $\xi_{j+1}^{(b)}$ to $\mathbf{FL}_\theta\xi_j^{(b)}$ with probability $A(\mathbf{FL}_\theta\xi_j^{(b)}|\xi_j^{(b)})$.
        **end for**
        $\mathcal{L}_{\text{Decoder}} \leftarrow \mathcal{L}_{\text{Decoder}} + \log p(x^{(b)}|z_J^{(s)}; \phi)$          ▷ With $\xi_J^{(b)} = \{z_J^{(b)}, v_J^{(b)}, d_J^{(b)}\}$
    **end for**
    $\phi \leftarrow \phi + \alpha_t \nabla_\phi \mathcal{L}_{\text{Decoder}}$
    $\psi \leftarrow \psi + \alpha_t \nabla_\psi \mathcal{L}_{\text{ELBO}}$
    $\theta \leftarrow \theta + \alpha_t \nabla_\theta \mathcal{L}_{\text{Sampler}}$
**end for**

---

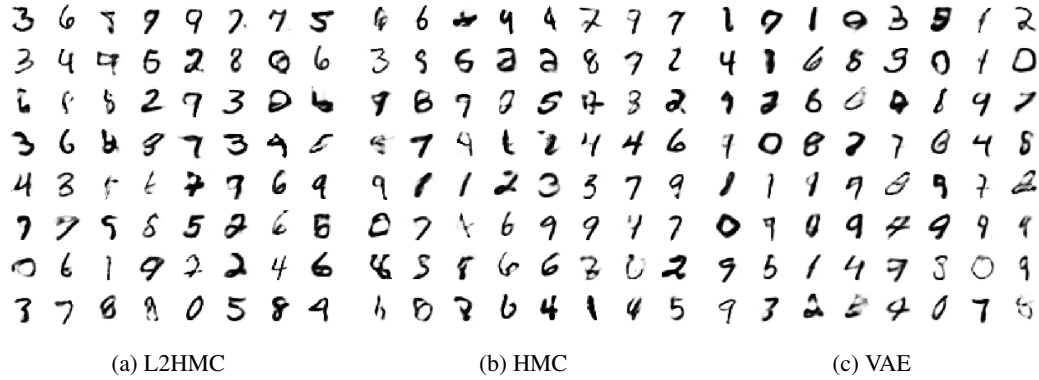

(a) L2HMC                    (b) HMC                    (c) VAE

Figure 5: L2HMC-DGLM decoder produces sharper mean activations.

networks $\{\cdot\}_x$ and $\{\cdot\}_v$. The architecture then consists of 2 hidden layers of 200 units and ReLU non-linearities. For $\lambda$ (scale parameter of the loss), we use the standard deviation of the approximate posterior.

**AIS Evaluation** For each data point, we run 20 Markov Chains in parallel, $10,000$ annealing steps with 10 leapfrogs per step and choose the step size for an acceptance rate of $0.65$.

### D.3 MNIST SAMPLES

We show in Figure 5 samples from the three evaluated models: VAE (Kingma & Welling, 2013), HMC-DGLM (Hoffman, 2017) and L2HMC-DGLM.

