# OpenReview forum: "Generalizing Hamiltonian Monte Carlo with Neural Networks"
_ICLR.cc/2018/Conference — Accept (Poster)_

### Official Review · AnonReviewer1 · 2017-11-26
**Nice idea; but the paper could be improved.**

**Rating:** 7
**Confidence:** 4

**Review:**

In this work, the authors propose a procedure for tuning the parameters of an HMC algorithm (I guess, if I have understood correctly).

I think this paper has a good and strong point: this work points out the difficulties in choosing properly the parameters in a HMC method (such as the step and the number of iterations in the leapfrog integrator, for instance). In the literature, specially in machine learning, there is ``fever’’ about HMC, in my opinion, partially unjustified.

If I have understood, your method is an adaptive HMC algorithm  where the parameters are updated online; or is the training  done in advance? Please, remark and clarify this point.

However, I have other additional comments:

- Eqs. (4) and (5) are quite complicated; I think a running toy example can help the interested reader.

- I suggest to compare the proposed method to other efficient methods that do not use the gradient information (in some cases as multimodal posteriors, the use of the gradient information can be counter-productive for sampling purposes), such as Multiple Try Metropolis (MTM) schemes

L. Martino, J. Read, On the flexibility of the design of Multiple Try Metropolis schemes, Computational Statistics, Volume 28, Issue 6, Pages: 2797-2823, 2013,

adaptive techniques,

H. Haario, E. Saksman, and J. Tamminen. An adaptive Metropolis algorithm. Bernoulli, 7(2):223–242, April 2001,

and component-wise strategies as Gibbs Sampling,

W. R. Gilks and P. Wild, Adaptive rejection sampling for Gibbs sampling, Appl. Statist., vol. 41, no. 2, pp. 337–348, 199.

At least, add a brief paragraph in the introduction citing and discussing this possible alternatives.

---

> ### Author Response · Authors · 2017-12-20
> **Adaptive techniques can be complementary to our method**
>
> Thank you for your review and the pointer to references.
>
> We wish to emphasize that our method is able, but not limited to, automatically tuning HMC parameters (which systems like Stan already have well-tested heuristics for). Our approach generalizes HMC, and is capable of learning proposal distributions that do not correspond to any tuned HMC proposal (but which can still be plugged into the Metropolis-Hastings algorithm to generate a valid MCMC algorithm). Indeed, in our experiments, we find that our approach significantly outperforms well-tuned HMC kernels.
>
> The training is done during the burn-in phase, and the trained sampler is then frozen. This is a common approach to adapting transition-kernel hyperparameters in the MCMC literature.
>
> Regarding the references, we added those in the text. We also want to emphasize that all of these are complementary to and could be combined with our method. For example, we could incorporate the intuition behind MTM by having several parametric operators and training each one when used.
>
> Additionally, in the process of revisiting our experiments to compare against LAHMC, we empirically found that weighting the second term of our loss (the ‘burn-in’ term) could lead to even more improved auto-correlation and ESS on the diagnostic distributions. We therefore updated the paper and report the results obtained with slightly tuning that parameter (setting it to 0 or 1).

---

### Official Review · AnonReviewer2 · 2017-11-27
**The paper proposed a generalized HMC with neural networks, but it lacks theoretical justifications why it could mix with multiple modes**

**Rating:** 6
**Confidence:** 3

**Review:**

The paper proposed a generalized HMC by modifying the leapfrog integrator using neural networks to make the sampler to converge and mix quickly. Mixing is one of the most challenge problems for a MCMC sampler, particularly when there are many modes in a distribution. The derivations look correct to me. In the experiments, the proposed algorithm was compared to other methods, e.g., A-NICE-MC and HMC. It showed that the proposed method could mix between the modes in the posterior. Although the method could mix well when applied to those particular experiments, it lacks theoretical justifications why the method could mix well.

---

> ### Author Response · Authors · 2017-12-20
> **Expected Square Jump Distance is related to mixing speed**
>
> Thank you very much for your review and comments. Guaranteeing mixing between modes is a fundamental (#P-Hard) problem. As such, we do not hope to solve it in the general case. Rather, we propose a method to greatly increase the flexibility and adaptability of a class of samplers which is already state of the art in many contexts. The relation between mixing time and expected square jump distance is thoroughly treated in [Pasarica & Gelman, 2010], and is the theoretical inspiration for our choice of training loss.
>
> We further emphasize that, barring optimization issues, our method should always fare at least as well as HMC in terms of mixing.
>
> Thank you once again, we have updated the text to more clearly discuss why our approach might be expected to lead to better mixing.
>
> Additionally, in the process of revisiting our experiments to compare against LAHMC, we empirically found that weighting the second term of our loss (the ‘burn-in’ term) could lead to even more improved auto-correlation and ESS on the diagnostic distributions. We therefore updated the paper and report the results obtained with slightly tuning that parameter (setting it to 0 or 1).

---

### Official Review · AnonReviewer3 · 2017-11-27
**Good paper on fast mixing extension of HMC**

**Rating:** 8
**Confidence:** 2

**Review:**

The paper introduces a non-volume-preserving generalization of HMC whose transitions are determined by a set of neural network functions. These functions are trained to maximize expected squared jump distance.
This works because each variable (of the state space) is modified in turn, so that the resulting update is invertible, with a tractable transformation inspired by Dinh et al 2016.

Overall, I believe this paper is of good quality, clearly and carefully written, and potentially accelerates mixing in a state-of-the-art MCMC method, HMC, in many practical cases. A few downsides are commented on below.

The experimental section proves the usefulness of the method on a range of relevant test cases; in addition, an application to a latent variable model is provided sec5.2.
Fig 1a presents results in terms of numbers of gradient evaluations, but I couldn't find much in the way of computational cost of L2HMC in the paper. I can't see where the number "124x" in sec 5.1 stems from. As a user, I would be interested in the typical computational cost of both "MCMC sampler training" and MCMC sampler usage (inference?), compared to competing methods. This is admittedly hard to quantify objectively, but just an order of magnitude would be helpful for orientation.
Would it be relevant, in sec5.1, to compare to other methods than just HMC, eg LAHMC?

I am missing an intuition for several things: eq7, the time encoding defined in Appendix C

Appendix Fig5, I cannot quite see how the caption claim is supported by the figure (just hardly for VAE, but not for HMC).

The number "124x ESS" in sec5.1 seems at odds with the number in the abstract, "50x".

# Minor errors
- sec1: "The sampler is trained to minimize a variation": should be maximize
"as well as on a the real-world"
- sec3.2 "and 1/2 v^T v the kinetic": "energy" missing
- sec4: the acronym L2HMC is not expanded anywhere in the paper
The sentence "We will denote the complete augmented...p(d)" might be moved to after "from a uniform distribution" in the same paragraph.
In paragraph starting "We now update x":
    - specify for clarity: "the first update, which yields x' "/ "the second update, which yields x''  "
    - "only affects $x_{\bar{m}^t}$": should be $x'_{\bar{m}^t}$  (prime missing)
    - the syntax using subscript m^t is confusing to read; wouldn't it be clearer to write this as a function, eg "mask(x',m^t)"?
    - inside zeta_2 and zeta_3, do you not mean $m^t" and $\bar{m}^t$ ?
- sec5: add reference for first mention of "A NICE MC"
- Appendix A:
    - "Let's" -> "Let"
    - eq12 should be x''=...
- Appendix C: space missing after "Section 5.1"
- Appendix D1: "In this section is presented" : sounds odd
- Appendix D3: presumably this should consist of the figure 5 ? Maybe specify.

---

> ### Author Response · Authors · 2017-12-20
> **Clarifications and comparison with LAHMC**
>
> We first and foremost want to thank you for your time and extremely valuable comments. We have uploaded a new version of the paper based on the feedback, and have addressed specific points below.
>
> Clarification about 50x vs 124x:
> We decided against advertising the 124x number as it is misleading considering that HMC completely failed on this task; the correct ratio was too large for us to experimentally measure. As such, we reported the one for the Strongly-Correlated Gaussian. We clarified this in the text and detail that L2HMC can succeed when HMC fails.
>
> Intuition on Eq 7.:
> We define this reciprocal loss to encourage mixing across the entire state space. The second term corresponds exactly to Expected Square Jump Distance, which we want to maximize as a proxy for mixing. The first term discourages a particle from not-moving at all in a region of state space -- if d(x, x’) = 0, the first term would be infinite. We clarified that part in the text.
>
> Time encoding:
> Our operator L_\theta consists of the composition of M augmented leapfrog steps. For each of those leapfrog, the timestep t is provided as input to the networks Q, S and T. Instead of providing it as a single scalar value, we provide it as a 2-d vector [cos(2 * pi * t / M), sin(2 * pi * t / M)].
>
> Regarding samples in Fig5:
> Sample quality and sharpness are inherently hard things to evaluate. Our observation was that many digits generated by L2HMC-DGLM look very sharp (Line 1 Column 2, Line 2 Column 8, Line 5 Column 2, Line 7 Columns 3 and 7…). However, we will weaken the claim in the caption.
>
> Comparison with LAHMC:
> We compared our method to LAHMC on the evaluated energy functions. L2HMC significantly outperforms LAHMC on all tasks, for the same number of gradient evaluations. LAHMC is also unable to mix between modes in the MoG case. Results are reported in Appendix C.1.
>
> We also note that L2HMC could be easily combined with LAHMC, by replacing the leapfrog integrator of LAHMC with the learned one of L2HMC.
>
> In the process of revisiting our experiments to compare against LAHMC, we empirically found that weighting the second term of our loss (the ‘burn-in’ term) could lead to even more improved auto-correlation and ESS on the diagnostic distributions. We therefore updated the paper and report the results obtained with slightly tuning that parameter (setting it to 0 or 1).
>
> Question about computation:
> For the 2d-SCG case, on CPU, the training of the sampler took 160 seconds. The L2HMC overhead for sampling, with a batch-size of 200, was about 36%. This is negligible compared to an 106x improved ESS.  We also should note that for the latent generative model case, we train the sampler online with the same computations used to train everything else; in that case L2HMC and HMC perform the exact same number of gradient evaluation of the energy and thus requires no training budget.
>
> Thank you once again for your valuable feedback, we hope this helps answer your questions!

---

### Public Comment · (anonymous) · 2018-01-02
**Question regarding Section 4.1.2**

I had one question -- in Equations 4-6, you have functions Q, T to rescale and translate the momentum and position. However it seems that Q, T are vectors and thus you are learning arbitrary transformations to d_x U(x)?

If that is the case, then I'm unclear on how your leapfrog operator guarantees (approximate) integration of the Hamiltonian. And if it does not and your goal is simply to learn proposals for which you can compute the Jacobian, then what's the purpose of the momentum resampling step and/or having the d_x U(x) term in the update at all?

If you could shed some light on that, that would be great!

---

> ### Author Response · Authors · 2018-01-04
> **Response**
>
> Thank you for your question.
>
> You are correct that in general our method’s proposals cannot be interpreted as (approximately) integrating the dynamics of any Hamiltonian. Ultimately our goal (and that of HMC) is to produce a proposal that mixes efficiently, not to simulate Hamiltonian dynamics accurately.
>
> There are many other trainable proposals for which we could compute the Jacobian, but not all will mix efficiently. By choosing a parameterized family of proposals that can mimic the behavior of HMC (and initializing it to do so), we ensure that our learned proposal performs at least as well as HMC.
>
> The momentum-resampling step is essential, since it is the only source of randomness in the proposal. Using gradient information (d_x U(x)) is essential for giving the proposal information about the local geometry of the target distribution.

---

> > ### Public Comment · (anonymous) · 2018-01-05
> > **Re:**
> >
> > Thanks for the response! As I understand it then, your method is the first in literature to be able to train expressive MCMC kernels? (as if I recall correctly, in the past, the focus has been more on tuning a very limited number of parameters associated with the proposal distribution, like the variance of a gaussian proposal for ex.)

---

> > > ### Author Response · Authors · 2018-01-09
> > > **Re:**
> > >
> > > We believe we are the most flexible parameterization of a Markov kernel to date. However, there has been previous work that proposes general purpose kernels. Most relevant is Song et al, which trains a flexible class of volume-constrained Markov proposals using an adversarial objective. (We discuss and experimentally compare against this approach in our paper.)
> > >
> > > Thanks for the question!

---

### Author Response · Authors · 2018-01-04
**Updated paper**

We thank the reviewers for their valuable time and comments.

We updated the paper with the following modifications:
- Clarified some points and fixed typos pointed out by the reviewers.
- Added a ``Future Work section as well as additional relevant references.
- Added a comparison with Look-Ahead HMC (LAHMC; Sohl-Dickstein et al. 2014) in the Appendix.

Additionally, in the process of revisiting our experiments to compare against LAHMC, we empirically found that weighting the second term of our loss (the ‘burn-in’ term) could lead to even more improved auto-correlation and ESS on the diagnostic distributions. We therefore updated the paper and report the results obtained with slightly tuning that parameter (setting it to 0 or 1).

---

### Public Comment · (anonymous) · 2018-01-15
**How is the discontinuity in A(.|.) handled?**

The accept-reject step of the MCMC kernel introduces a discontinuity in the function A( ) that depends on both the random variable xi AND the parameter being optimized with respect to. This means that interchanging the order of expectation (integration) and differentiation in eq. (8) is invalid in general. Have the authors considered this in their derivations? Can you prove that the gradients used for learning in Alg. 1 are still unbiased, and thus will lead to convergence by standard stochastic approximation results? If the gradients are not unbiased (which I suspect is the case), have you studied the impact this has on the learning procedure?

---

> ### Author Response · Authors · 2018-01-15
> **We believe our derivations are correct and the gradients unbiased**
>
> Thank you for your question, we believe our derivations are correct and the gradients unbiased.
>
> In Eq. (8), p(\xi) and q(\xi) are not functions of the parameters \theta and the loss inside the expectation is an (almost everywhere) differentiable function of \xi. This allows us, when differentiating w.r.t \theta to easily exchange (under mild assumptions) derivative and integration.
>
> It is important to note that when optimizing, we are NOT sampling through the accept/reject step. Given a state \xi, we move it forward using our (differentiable) generalized Hamiltonian dynamics and use that new proposed state for the loss; explicitly marginalizing over the accept/reject decision. We thus do not need to back-propagate through a discrete decision variable, making our gradients unbiased. This is additionally detailed in Pasarica and Gelman 2010.
>
> We hope this answers your question!

---

> > ### Public Comment · (anonymous) · 2018-01-18
> > **Thanks for the prompt response**
> >
> > I think the key issue here is establishing whether the integrand in Eq. (8) is an absolutely continuous function of \theta for almost all \xi. Then you can use e.g. Theorem 3 here http://planetmath.org/differentiationundertheintegralsign to validate the interchange. The easier to validate Theorem 2, which is sufficient for most cases in stochastic gradient-based VI, does not hold for your integrand because assumption 2 is not valid for the function A(|). This because the derivative of A(|) does not exist whenever the ratio of densities is exactly one in Eq. (3). But perhaps it is easy to show the absolute continuous property of A(|) wrt \theta for almost all \xi?
> >
> > I do agree that this is not an issue of any discrete random variables nor the function described by the numerical integration of an Hamiltonian flow. My concern is merely if the discontinuity in the gradient of A(|) wrt \theta will be an issue.
> >
> > Also, thanks for a very interesting read!

---

> > > ### Author Response · Authors · 2018-01-22
> > > **A sub-gradient exists everywhere**
> > >
> > > While it is true that for some value of \xi, the integrand is not differentiable, it does admit a sub-gradient everywhere. This is sufficient for optimization ( https://en.wikipedia.org/wiki/Subgradient_method ). Also note that ReLU networks demonstrate this same characteristic (continuous function, with discontinuities in the first derivative), but are routinely trained in deep learning.
> > >
> > > Thank you again for your interest and thoughtful reading of our work!

---

> > > > ### Public Comment · (anonymous) · 2018-01-24
> > > > **Interesting point**
> > > >
> > > > Yes, that is an interesting point that ReLU networks are routinely used together with stochastic gradient VI. My concern would then apply to these methods as well, even though the discontinuity in the MH method is inherent to the problem where for NNs it can be resolved by replacing ReLUs with a continuously differentiable alternative.
> > > >
> > > > It is true that sub-gradients exists, but I believe the interaction between sub-gradients and integration requires some careful consideration.
> > > >
> > > > I think the paper is good work and my comments are mostly me being curious if you had considered this problem (and whether it is a problem) and had any ideas how to resolve it. Thanks for the discussion!

---

### Decision · Program_Chairs · 2018-01-29
**ICLR 2018 Conference Acceptance Decision**

**Decision:**

Accept (Poster)

**Comment:**

This paper presents a learned inference architecture which generalizes HMC. It defines a parameterized family of MCMC transition operators which share the volume preserving structure of HMC updates, which allows the acceptance ratio to be computed efficiently. Experiments show that the learned operators are able to mix significantly faster on some simple toy examples, and evidence is presented that it can improve posterior inference for a deep latent variable model. This paper has not quite demonstrated usefulness of the method, but it is still a good proof of concept for adaptive extensions of HMC.